# Evaluation of Radio Frequency Identification Power and Unmanned Aerial Vehicle Altitude in Plant Inventory Applications

**Van Patiluna [1,2], Joe Mari Maja [1,*] and James Robbins [3]**

[1] Center of Applied Artificial Intelligence for Sustainable Agriculture, 1890 Research and Extension, South Carolina State University, 300 College Ave., Orangeburg, SC 29117, USA; vpatilun@scsu.edu

[2] Department of Computer Engineering, University of San Carlos, Cebu City 6000, Philippines

[3] Ornamental Hort Solutions, 3923 Oakwood Road Apt1, Little Rock, AR 72205, USA; jrobbins4930@gmail.com

**\*** Correspondence: jmaja@scsu.edu

**Abstract:** In the business of growing and selling ornamental plants, it is important to keep track of plants from nursery to distribution. Radio Frequency Identification (RFID) technology provides an easier tracking method for inventories of plants by attaching tags with unique identifiers. Due to the vast area of most nurseries, there is a need to have an efficient method of scanning RFID tags. This paper investigates the use of drones and RFID, specifically, the effects of RFID reader power and flight altitude on tag counts. The experimental setup evaluated three RFID reader power levels (15 dBm, 20 dBm, and 27 dBm), three flight altitudes (3 m, 5 m, and 7 m), the number of passes (one or two), and two plant types ('Green Giant' arborvitae and 'Sky Pencil' holly). For RFID tags, four types were used (L5, L6, L8, and L9), with two antenna types (dog-bone and square-wave) and two attachment types (loop-lock and stake). For each power level, the UAV was flown to three different altitudes of 3 m, 5 m, and 7 m above the ground. At each altitude, two scan passes were performed at a constant speed of approximately 1.5 m/s. Each plot of plants (two in total) was randomly tagged with a total of 40 RFID tags per plot. Field data were collected from September to December 2023 (on a total of eight dates). The data showed that a power level of 15 dBm and an altitude of 3 m yielded a tag count of 53%, while counts of 34% and 16% were achieved at 5 m and 7 m, respectively. At 20 dBm and an altitude of 3 m, the count accuracy across all tag types and both plants was 90%. When the altitude was increased to 5 m and 7 m, tag-count accuracy dropped to 75% and 33%, respectively. The highest count accuracy was observed at 27 dBm and an altitude of 3 m, with a reading accuracy of 98%. Tag types L6 and L9 performed better at any power level and altitude, while L5 and L8 performed well at a higher power level and lower altitude. In this experiment, canopy properties (size and shape) had no effect on the number of tags read. This study aimed to evaluate the RFID power and UAV altitude achieving the highest accuracy in scanning the RFID tags. Furthermore, it also assessed the effects of plant growth on the scanning efficiency and accuracy of the system.

**Keywords:** UAV; RFID; plant inventory

## 1. Introduction

The ornamental horticulture industry in the United States is a multibillion-dollar industry that significantly contributes to the economy of every state [1]. The economic impact of the green industry, including ornamental plants, is substantial, with a total output of tree production and tree care services valued at $14.55 billion, translating into $21.02 billion in total output impacts, 259,224 jobs, and $14.12 billion in value added [2]. The largest individual industry sectors in terms of employment and GDP contributions are

landscaping and horticultural services; greenhouse, nursery, and floriculture production; and lawn and garden equipment and supplies stores [3]. The production and commercialization of ornamental plants have faced challenges, particularly due to the impact of the COVID-19 pandemic. Substantial changes have occurred in food and ornamental plant production chains, and variation in the size of companies between small and large producers is common in the flower and ornamental plant production sector [4,5]. Despite these challenges, the ornamental plant agribusiness has shown increasing production trends, indicating its resilience and potential for growth [6]. The significance of ornamental plants extends beyond their economic value. Ornamental plants play a crucial role in developing the agricultural sector and agro-tourism, contributing to the overall diversification and growth of the horticultural industry [7]. Additionally, the use of ornamental plants in phytoremediation presents an ecological alternative for removing contaminants from water, air, and soil, highlighting their environmental importance [8].

The significance of inventory in ornamental plant production is multifaceted and crucial for various aspects of the industry. Inventory management plays a pivotal role in ensuring the availability of diverse ornamental plant species, which is essential for meeting the demands of the market and consumers [9]. Additionally, inventory management is vital for maintaining the visual quality of plants and determining their productivity, which ultimately influences their commercial value [9]. Furthermore, inventories of ornamental plants are essential for assessing the attractiveness of these plants to natural enemies of pests, which can have implications for pest control in horticultural settings [10]. Moreover, inventory management is integral for the conservation and sustainable exploitation of ornamental plant species. Neglected and underutilized plants (NUPs) have the potential for sustainable exploitation in the ornamental horticultural sector, and inventorying these species is a crucial step in recognizing their value and feasibility for commercialization [11]. Additionally, the conservation of ornamental plant genetic resources through inventory management is essential for preserving biodiversity and supporting research initiatives [12].

Furthermore, inventory management is important for addressing challenges such as plant diseases associated with phytoplasma and viruses, which can significantly impact the production of ornamental plants and lead to economic losses [13]. By maintaining accurate inventories, producers can effectively monitor and manage the health of ornamental plants, mitigating the impact of diseases and ensuring the quality of the plants for commercial purposes.

Using drones in inventory management for ornamental plant production has emerged as a significant technological advancement with many implications. Drones, also known as unmanned aerial vehicles (UAVs), have been increasingly employed in agriculture for monitoring and managing crop production [14,15], and their application in the ornamental plant industry is gaining traction [16]. Drones serve as a smart farming technology, enabling the monitoring and prediction of crop performance and assessing the need for and impacts of fertilizer and pesticide applications. This technology has the potential to revolutionize inventory management in ornamental plant production by providing efficient and accurate data collection on plant growth, health, and spatial distribution. Integrating drones in inventory management for ornamental plants offers several advantages, including the ability to conduct rapid and comprehensive surveys of plant populations, leading to improved decision-making processes for producers [16]. Drones can facilitate the assessment of plant health. Additionally, using drones can contribute to optimizing resource utilization, such as of water and fertilizers, by providing real-time data on plant conditions and environmental factors. Furthermore, the employment of drones in inventory management aligns with the broader trend in precision agriculture, where technology is leveraged to enhance the efficiency and sustainability of agricultural practices [16]. The high-resolution imaging capabilities of drones enable detailed monitoring of ornamental plant nurseries, allowing for precise inventory assessments and identifying areas for improvement or intervention. This level of precision can lead to more targeted

and effective management strategies, ultimately contributing to the overall productivity and quality of ornamental plant production.

RFID and drones are increasingly being used for inventory management in various industries. The integration of RFID technology with drones has shown promising results in enhancing inventory control and supply chain management. Li et al. (2021) established a passive RFID localization scheme based on drones for inventory management in warehouses, demonstrating the potential of this technology in real-world applications [17]. Furthermore, Turkler et al. (2022) highlighted the use of drones carrying RFID readers for dynamic inventory checks in factory environments, emphasizing the practicality of this approach in industrial settings [18]. Additionally, Guruswamy et al. (2022) emphasized the implementation of RFID and drone technology in retail supply for tracking products and improving inventory management, further supporting the relevance of these technologies in supply chain operations [19]. Moreover, Wang et al. (2017) estimated RFID values from the perspective of inventory management and proved that RFID revolutionized supply chain management [20]. Additionally, Li and Visich (2006) mentioned that RFID can be used to track the movement of products through the supply chain from production to the retail point of sale in real time, providing higher visibility for inventory and assets in the supply chain [21]. With RFID systems, companies would have increased product visibility, reduce out-of-stock items, trim warehouse costs, eliminate stock errors, reduce theft and shrinkage, and allow companies to regularly update their logistics and inventory databases. Furthermore, Li et al. (2022) investigated drone-based RFID localization for fast and accurate inventory management [17]. The literature has also discussed the potential of drones to provide valuable information for inventory tracking and management in various industries, including construction projects. Quino et al. (2021) used drones and RFID to address the specific need to move toward on-demand plant inventory [22]. Their work focused on evaluating different RFID tags with respect to the tags' distance and orientation in relation to the RFID reader. They also developed their own RFID reader, which could be attached to a drone. Quino et al. (2022) also investigated the relationship between drone speed and the number of flights needed to increase the number of scans of RFID tags made by the drone [23]. Cutting-edge drone technology has been identified to make inventory control more economical and efficient, aligning with the growing interest in leveraging drones for supply chain optimization. This could be further expanded into multiple drones flying together in a swarm [24], increasing efficiency even more.

Although outdoor plant inventorying using RIFD and UAVs has been investigated, there has been no definitive result concerning the system's accuracy in reading the tags attached to the plants. For plant inventory applications, accuracy in scanning and counting the tags is paramount. This study aims to evaluate the RFID power and UAV altitude achieving the highest accuracy in scanning the RFID tags. Furthermore, it also assesses the effects of plant growth on the scanning efficiency and accuracy of the system.

## 2. Materials and Methods

The present study conducted experiments in reading RFID tags attached to plants in a nursery via an RFID-RM mounted on a UAV at different altitudes and RFID power settings.

### 2.1. Study Site, UAV, and RFID Tags

The field data were collected at Dudley Nurseries in Thomson, GA, USA (33.52242, −82.51449). A heavy lift drone (Matrice 600 Pro, Shenzhen, China) (Figure 1a) was used to carry the 285 g RFID Reader Module (RFID-RM) (Figure 1b) and antenna. The RFID-RM and antenna were used to scan the RFID tags [22] that were attached to plants' stems or stakes [23]. The UAV was able to handle a maximum payload of 6 kg, including a single Turnigy 2200 mAh LiPo battery.

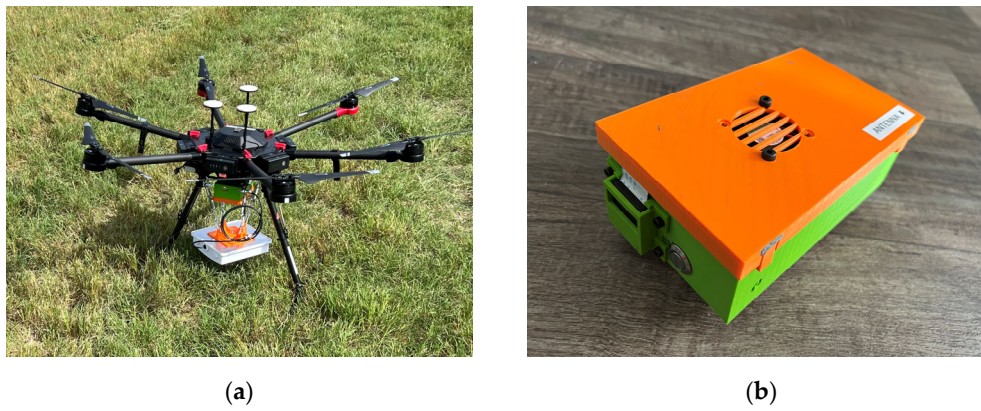

(**a**)                                                                (**b**)

**Figure 1.** (**a**) DJI Matrice 600 Pro and (**b**) RFID Reader Module (RFID-RM).

Figure 2a shows the RFID tags used in the experiment. All the tags used in the experiment were passive and operated at a frequency of 900 Mhz. There were four types with two different antenna designs (dog-bone and square-wave) and two attachment mechanisms (loop-lock and stake). A summary of the tag types is shown in Table 1. The tags were manufactured by Avery Dennison Corp., Mentor, OH, USA. The experimental layout also included 2 marker tags (Figure 2b) of the L5 type, which were used to identify the number of scan passes and the UAV height, respectively. The markers allowed us to automate the data collection without having to land the UAV every time the height was changed, which minimized the experiment time by 67%, from 9 take-off and landing cycles to just 3. The previous method required the UAV to land after each flight level and power setting combination. The Electronic Product Codes (EPCs) of the markers were used to separate the data in the log file into data per pass and UAV altitude.

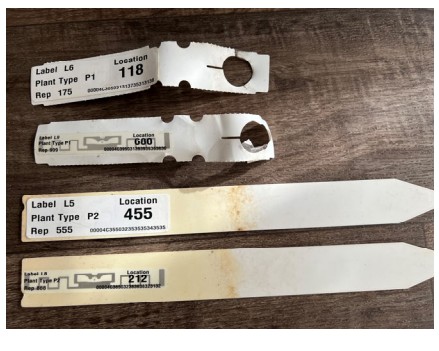
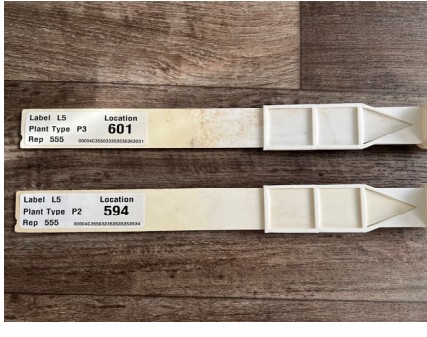

(**a**)                                                                (**b**)

**Figure 2.** RFID tag types (**a**) and marker tags (**b**).

**Table 1.** Summary of RFID tag types.

| Tag Type | Antenna | Attachment |
|----------|---------|------------|
| L5 | dog-bone | stake |
| L6 | dog-bone | loop-lock |
| L8 | square-wave | stake |
| L9 | square-wave | loop-lock |

There were 20 tags per type spread across the two plots. Each plot included a total of 40 tags, with 10 tags of each type. The 40 tags were placed randomly within each plot. The location for each tag was determined using a randomizer program.

### 2.2. RFID Reader Module (RFID-RM) and Dashboard Application

The RFID module used in the experiment was smaller and lighter than the one used in prior work [22]. The RFID-RM utilized an ARM microcontroller (ATSAM3X8E, Microchip, Chandler, AZ, USA) and an RFID Module chip (M6E-NANO, Novanta, MA, USA). The RFID chip could be programmed to output a maximum power of 27 dBm. It also had a transceiver (xBee, Digi, MN, USA), real-time clock (R.T.C.), and microSD card. The antenna used in this experiment was also lighter, although it had the same specifications as the antenna used in a previous experiment [22]. The antenna was suspended underneath the RFID-RM (Figure 3) with a carabiner and tensioner, unlike the previous experiment, in which the antenna was rigidly attached. The current configuration allowed the antenna to sway and be detached quickly from the UAV. The module was powered by a 2200 mAh 4-cell LiPo battery mounted on the UAV chassis.

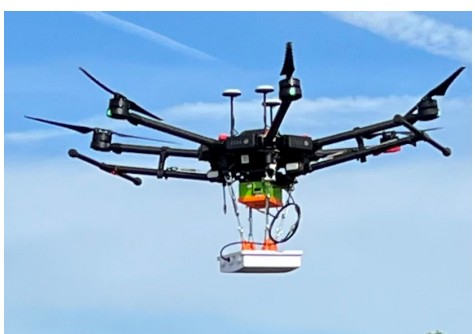

**Figure 3.** RFID-RM and suspended antenna.

The microcontroller in the RFID-RM controlled the RFID chip by setting the power output and turned on the antenna to start scanning. The antenna emitted radio frequency signals where the passive tags were energized then transmitted the data back to the antenna. Data read by the RFID chip were sent serially to the microcontroller, where they were stored to the microSD card and simultaneously broadcast wirelessly to the base computer via the xBee transceiver.

A dashboard application installed in the base computer (Figure 4a) was developed to control the RFID power setting and record the tag IDs scanned by the RFID-RM. Additionally, the dashboard application was able to enable/disable saving data to the microSD card and set the tag EPCs for the 'pass' and 'height' marker tags. To minimize delays in data reception from the RFID-RM, the dashboard application did not print the tag EPCs. Another application (Figure 4b) was developed to display the tag EPCs and monitor the tag EPCs for the 'pass' and 'height' tag EPCs. Both applications ran side by side, and each received data from its own xBee transceiver.

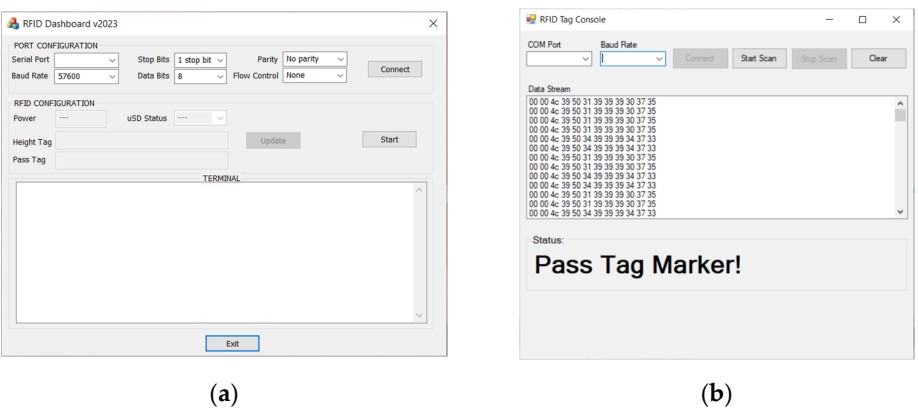

(**a**)           (**b**)

**Figure 4.** The (**a**) main dashboard and (**b**) marker detection applications.

### 2.3. Plant Type and Plot Layout

Two plant types were provided by Dudley Nurseries for this study: *Thuja X 'Green Giant'* (Figure 5a) and *Ilex crenata 'Sky Pencil'* (Figure 5b). 'Green Giant' arborvitae is an upright-growing needle evergreen, and 'Sky Pencil' holly is an upright-growing broadleaf evergreen. Production blocks of these plants were located in different areas of the nursery, namely block W7 (33.52218, −82.51450) for the 'Green Giant' and block W10 (33.52330, −82.51448) for the 'Sky Pencil.' Forty plants were randomly selected within the larger production block for tag treatments. The term 'plot' describes the sub-area within the larger nursery block where tagged plants were located. On each reading date (4 total), 10 plants were randomly selected for canopy measurements (height from the substrate surface to the uppermost foliage and canopy width measured in two directions at right angles). Figure 6a shows the aerial view of plot W7, and tag assignments, including the 'pass' and 'height' tags, are described in the assignment map in Figure 6b. Tagged plants were one row and column apart, with the pass and height markers situated on the front side (Figure 7). Plot W10 had a layout similar to plot W7; however, the tag placements were different due to the random tag assignment.

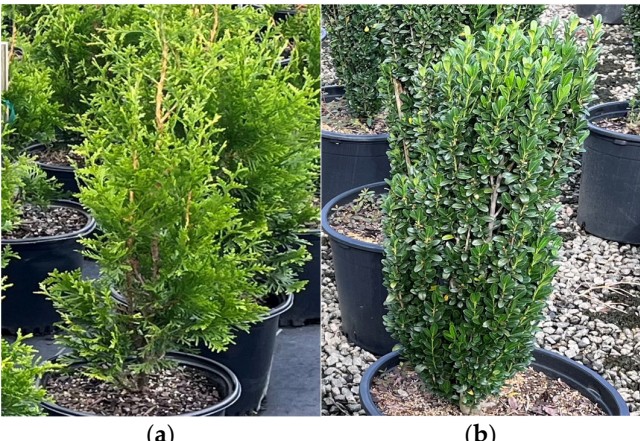

(**a**)  (**b**)

**Figure 5.** 'Green Giant' arborvitae (**a**) and 'Sky Pencil' holly (**b**).

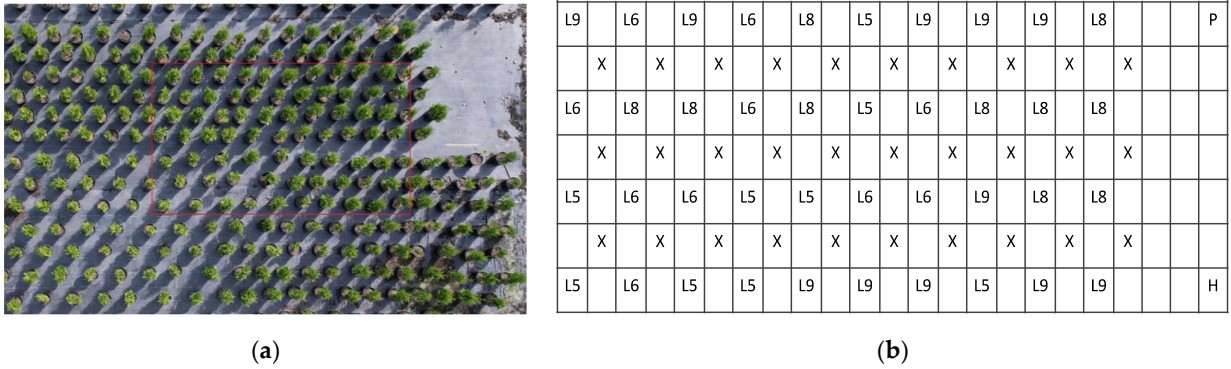

(**a**)  (**b**)

**Figure 6.** W7 study area (**a**) and RFID tag assignment map (**b**).

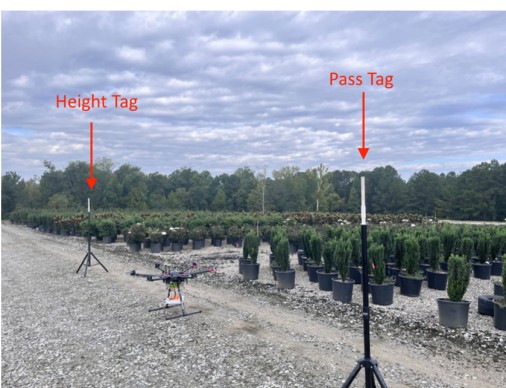

**Figure 7.** Pass and height tag positions relative to the plot.

### 2.4. UAV Flight Plan and Repetitions

The experimental design evaluated three RFID power settings: 15 dBm, 20 dBm, and 27 dBm. The UAV was flown at three altitudes for each power setting: 3 m, 5 m, and 7 m. The flight plan for each power setting was as follows: The UAV would take off and fly to the lowest altitude (3 m), where it would make two passes in a U-shaped pattern. Between passes at the same altitude, the UAV would fly to the pass marker once the 'pass' EPC tag was acknowledged by the marker detection application. Once the second pass was completed, the UAV was flown over the height tag marker until the 'height' EPC tag was detected. The UAV would be flown up to the next altitude (5 m) and the process repeated until completion of all three altitudes. Once flights had been completed at all three altitudes the UAV had landed, the RFID-RM was set to the next power level. Landing the UAV was required since the RFID power setting could not be re-programmed midflight. Once the power level was changed, the entire process was repeated two more times, as described above, until a total of three take-off and landing cycles were conducted per plot. The ground speed of the UAV was maintained at approximately 1.5 m/s.

Before any UAV flight campaign, a scan of the plot was performed using a handheld RFID reader (RFD8500, Zebra, Lincolnshire, IL, USA) to ensure all the tags were present (40 unique tags per plot). The tag EPC and type for each plant were recorded in both plots. These data were used as a lookup table for tags that had been scanned and read. It allowed classification of the tags read and identified the scanning frequency of a certain tag type.

### 2.5. Data Processing

Since the RFID-RM module was able to scan all tags within its range, the logged data in the microSD card and dashboard application could have contained invalid or unknown RFID tags. Thus, a desktop application was created to process the data from the two main sources. Processing data from both sources allowed for cross-checking of the data for consistency, and it also acted as a redundancy system in case data from either source were corrupted. The data processing flow is described in Figure 8.

Log files were loaded to the data processing application as text files. The application read each datum (tag EPC) and identified its validity in the first 6 bytes, which should have been '00 00 4C'. Then the application looked for the EPCs of the 'pass' and 'height' markers and isolated the EPCs into each pass made per flight level. With the data sorted into passes per flight level, the EPCs in each pass were classified into their tag types (L5, L6, L8, or L9) by referencing them with a lookup table with the EPCs of the tags attached to the plants.

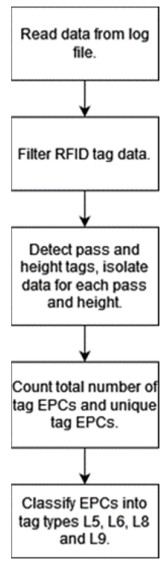

**Figure 8.** Data processing flow.

### 2.6. Plant Canopy Measurement

At each data collection date, the canopy sizes of 10 pre-selected plants for each species were measured to monitor growth. Three canopy measurements were taken, consisting of two width measurements (East–West, width 1 and North–South, width 2) and height from the substrate surface to the uppermost foliage. The 10 plants measured were randomly selected and tagged at the start of the experiment. The plants were measured before each flight campaign. A growth index (GI) was calculated for every measurement period using the following formula:

$$GI = \pi \ x \ h \ x \ r^2 \tag{1}$$

where:

$h$—plant height;

r—mean of width 1 and width 2.

Black plastic containers for 'Green Giant' arborvitae were size #3 (C-1200; 24 cm tall × top diameter of 28 cm), and those for 'Sky Pencil' holly were #7 (C-2800; 29 cm tall × top diameter of 36 cm). Both containers were manufactured by Nursery Supplies, Chambersburg, PA, USA. The size of the experimental plot within each larger nursery production block (W7 and W10) differed slightly due to the container spacing used by the nursery. Figure 9 shows the spacing and layout of the plants in both plots.

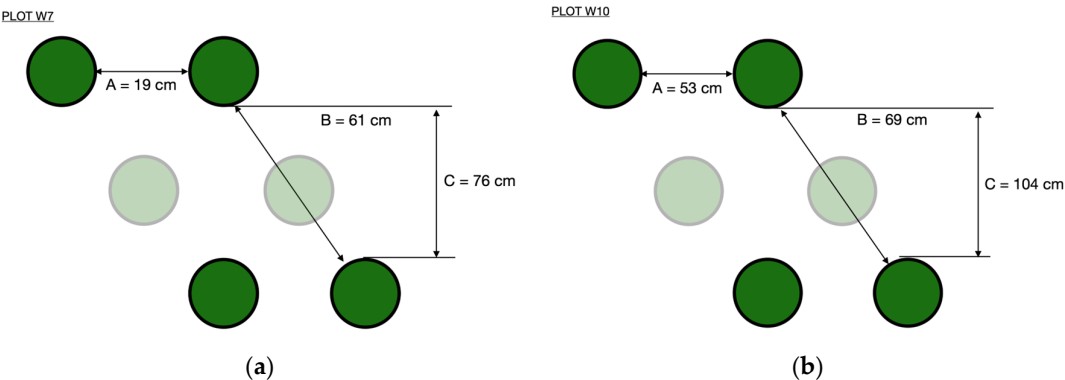

(**a**) (**b**)

**Figure 9.** Pot spacing measurements for blocks W7 (**a**) and W10 (**b**).

*2.7. Scanning Efficiency and Accuracy*

To determine the efficiency and accuracy of the tag-counting process, the net difference in unique tags read between the first and second passes (*diff*) was calculated. A negative value indicated a decrease in the number of unique tags detected between the first and second passes.

$$diff = pass2 - pass1 \tag{2}$$

where:

*pass1*—number of unique tags detected in the first pass;

*pass2*—number of unique tags detected in the second pass.

Count accuracy was determined by the number of unique valid tags read per pass at each RFID power setting and UAV altitude. This also constituted the sum of the unique tag per type. To achieve 100% accuracy, there should have been 10 unique tags detected per tag type.

$$Accuracy = \frac{L1 + L2 + L3 + L4}{40} \: x \: 100\% \tag{3}$$

*2.8. Statistical Analysis*

The plant growth index (GI) was correlated by the total number of RFID tags read before processing. The more RFID tags read by the system, the higher the chance of reading all the unique tags. This was important in achieving the highest count accuracy. Linear regression was applied to analyze the relationship between the plant growth index and the total number of tags counted. The plant GI was set as the independent variable, while the dependent variable was the total tags read. For plot W7, there were only three measurement periods (September 21, October 27, and November 1) while plot W10 had four (September 21, October 27, November 1, and December 14). The plant GI in these periods was correlated with the total tags read in the same period. The hypothesis was that plant canopy growth would not affect the scan count of the tags. There should have been no large positive linear association between plant growth and tag count.

## 3. Results

In total, eight field experiments were conducted on September 9 and 21–22, October 19, November 1–2 and 27, and December 14–15. However, data for plot W7 were not gathered on December 14 and 15 due to some plants accidentally being harvested by the nursery along with the RFID tags. Each experiment consisted of three take-off and landing cycles with an average flight time of 5.4 min.

*3.1. Unique Tag Counts*

Figures 10 and 11 show the unique RFID tags read at different RFID power settings and UAV altitudes classified into different tag types. Data are the average of all flights. Each tag type should have a maximum of 10 counts.

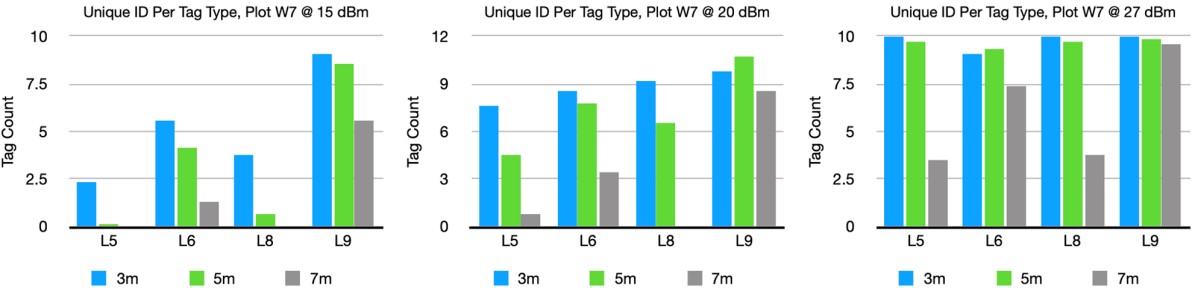

**Figure 10.** Average number of unique tags detected per tag type for plot W7.

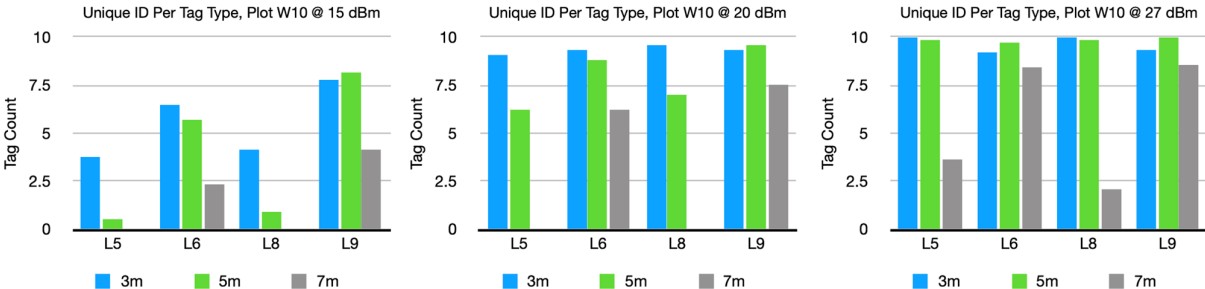

**Figure 11.** Average number of unique tags detected per tag type for plot W10.

Data were the average of the experiments. The lowest RFID power level of 15 dBm resulted in the lowest unique tag count, especially at higher altitudes. There was evidence of tags not being read, specifically for L5 at the altitude of 7 m. Increased RFID power improved the tag count but performed poorly, particularly with L5 and L8. Increasing the power to 27 dBm resulted in a significant increase in tag counts at lower altitudes. The highest tag count was achieved at 3 m, with 100% of tags scanned. Overall, tag types L6 and L9 resulted in the most unique tags detected at any power setting and UAV altitude. The performances of the tag types were due to the orientation of the antennas. The antennas for the loop-lock tags were oriented sideways, while stake tags were facing up. According to the study of Quino et. al. (2021), RFID tags that were oriented sideways produced the highest scan count [21].

### 3.2. Unique Tag Counts per Pass

Since the UAV performed two passes at a given altitude, it was important to know whether adding another pass improved the scanning performance. Figures 12 and 13 show the net difference (Equation (2)) in the number of tags scanned in the second pass. Data from September 9 were not included since the 'pass' marker tag was not implemented yet at that time.

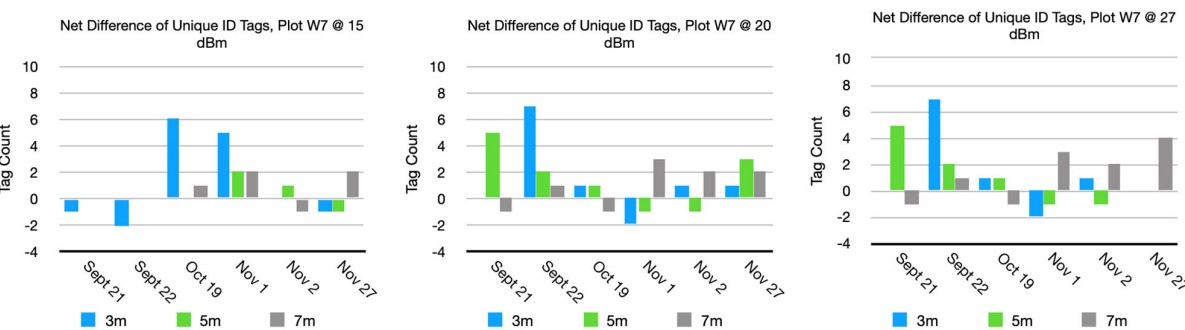

**Figure 12.** Net difference in the tags scanned after the second pass for plot W7.

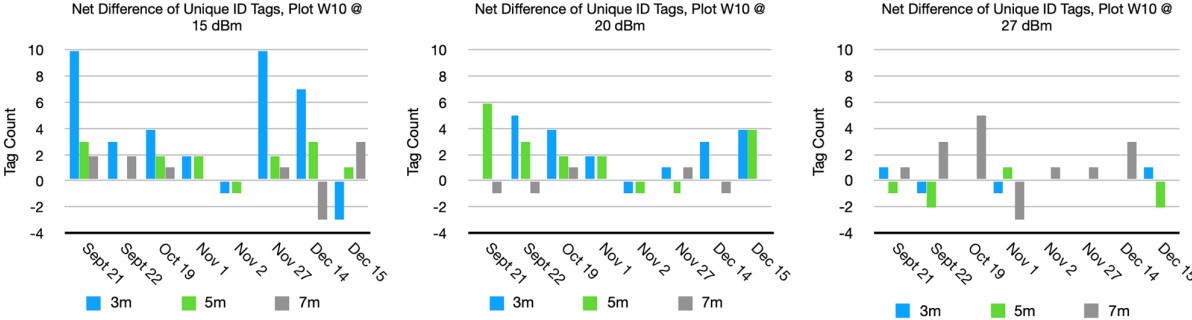

**Figure 13.** Net difference in the tags scanned after the second pass for plot W10.

For plot W7, 31% of the total passes had a negative difference, which means that the number of unique tags detected decreased on the second pass or there was a count loss. Plot W10 had a lower negative difference, at 22%. The data suggested that the first pass usually yielded a higher unique tag count than the second pass. Although the UAV was flown consistently, it was possible for it to veer from its flight path due to strong wind gusts that might have affected the count. Most of the tag-count losses happened at higher altitudes of 5 m and 7 m.

### 3.3. RFID Tag Count

Data observation from the previous experiments showed that data logged on the microSD card had more data and tags read than the log generated by the dashboard application. The logs from the dashboard application encountered invalid data due to communication errors. Although the higher power setting was efficient in detecting the tags, the increased power also allowed it to detect unknown tags. These unknown tags increased the data processing time. The trends of the total number of tags counted from the first experiment until the latest are shown in Figure 14. The data reflect the total tags counted at all UAV altitudes per RFID power setting. The counts exclude the 'pass' and 'height' marker tags.

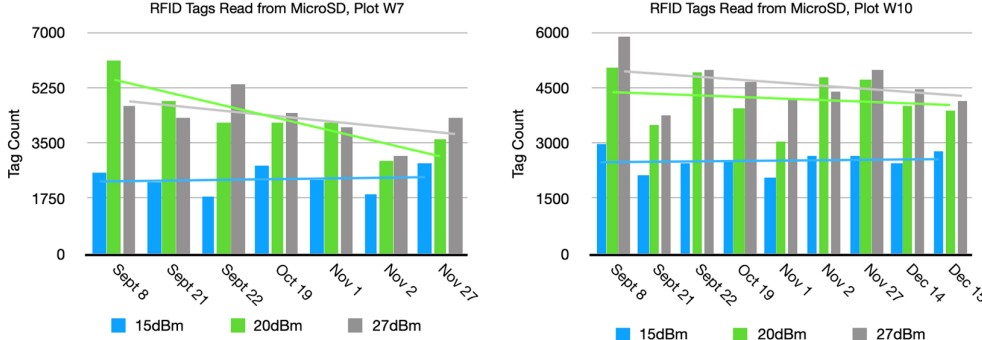

**Figure 14.** RFID tag data from microSD card for plots W7 and W10.

Based on the trendlines, it can be observed that the number of RFID tags counted was steady for the lowest power level at 15 dBm. At 20 dBm and 27 dBm, the tag counts declined, which was true for both plots. At the highest power level for plot W7, the trend showed that the number of tags read decreased until the final day. This unexpected increase in counts for plot W7 on the final day cannot be explained. For both plots, the trends suggested that there were no increases in the number of tags counted over the entire experimental period.

### 3.4. Plant Canopy Measurements

Figures 15 and 16 show the plants' height and width averages and growth indices in plots W7 and W10. Plants were measured on September 21, October 19, November 1, November 27, and December 14. Data from October 19 were not used in the overall average since they were later found to be in error. Plants in plot W7 ('Green Giant' arborvitae) showed slight growth, while plants in plot W10 ('Sky Pencil' holly) were steady. On December 14, some of the plants to be measured on plot W7 were no longer present as a result of being accidentally harvested by the nursery. For that reason, no plant measurements were collected for plants in plot W7 on that date.

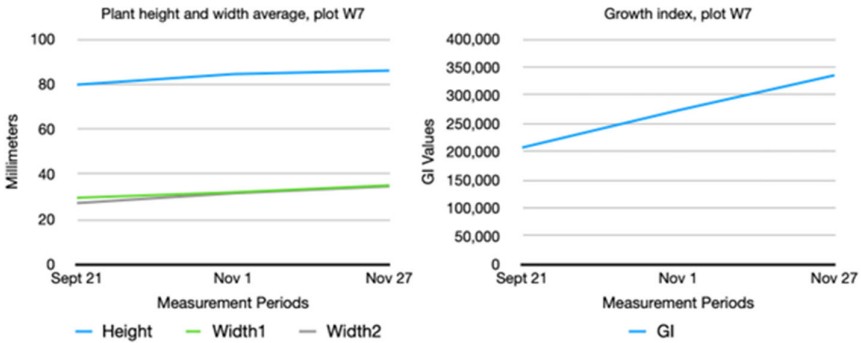

**Figure 15.** Plant measurement averages and growth indices for plot W7.

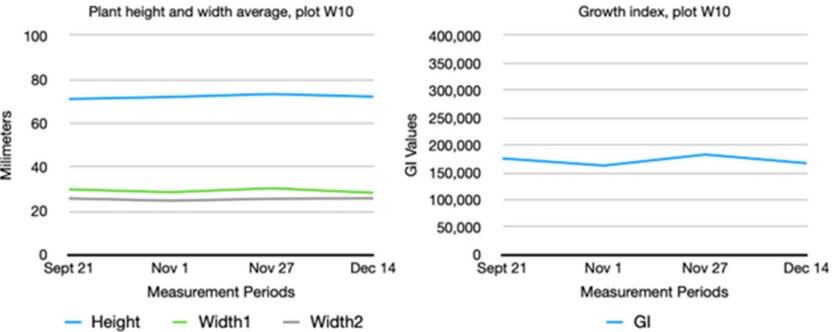

**Figure 16.** Plant measurement averages and growth indices for plot W10.

As observed, some of the width measurements fluctuated in plot W10, which is also reflected by the fluctuation in the growth indices. This was attributed to shrinkage in the container substrate and the measurement method of height and width. The assumption was that there would be significant growth during the period between September 21 and December 14. To determine if the plant growth was significant, the plant growth index (GI) and the elapsed days were correlated with September 21 as day 1, November 1 as day 41, November 27 as day 67, and December 14 as day 84. *p*-values were calculated as 0.07 and 0.90 for plots W7 and W10, respectively. Both *p*-values were above 0.05. Thus, the 84-day period was not statistically significant for plant growth; therefore, there was no significant plant growth.

### 3.5. Tag Scanning Accuracy

Tables 2 and 3 show the average scan count for each tag type in plots W7 and W10. The table shows that at the highest power setting of 27 dBm and the heights of 3 m and 5 m, the scanning accuracies (Equation (3)) were about 97% in plot W7 and 98% in plot W10. Although the accuracy was below 100%, the table shows that at the same power setting and height, most of the tag types had 100% detection (10/10). Since the data were averaged for all the experiments, some values were affected by the results of the previous experiments, where there were lower values. Table 4 shows the average of the data for plots W7 and W10.

**Table 2.** Scanning accuracy at different power settings and heights for plot W7.

| RFID Pwr | UAV Altitude | L5 | L6 | L8 | L9 | Total | Accuracy (%) |
|---|---|---|---|---|---|---|---|
| 15 dBm | 3 m | 2.25 | 5.58 | 3.83 | 9.08 | 20.75 | 52 |
|  | 5 m | 0.17 | 4.17 | 0.67 | 8.58 | 13.58 | 34 |
|  | 7 m | 0.00 | 1.25 | 0.00 | 5.58 | 6.83 | 17 |
| 20 dBm | 3 m | 7.58 | 8.50 | 9.17 | 9.83 | 35.08 | 88 |

| | 5 m | 4.58 | 7.75 | 6.58 | 10.67 | 29.58 | 74 |
| | 7 m | 0.75 | 3.42 | 0.08 | 8.50 | 12.75 | 32 |
| 27 dBm | 3 m | 10.00 | 9.08 | 10.00 | 10.00 | 39.08 | 98 |
| | 5 m | 9.75 | 9.33 | 9.83 | 9.92 | 38.83 | 97 |
| | 7 m | 3.50 | 7.42 | 3.83 | 9.58 | 24.33 | 61 |

**Table 3.** Scanning accuracy at different power settings and heights for plot W10.

| RFID Pwr | UAV Altitude | L5 | L6 | L8 | L9 | Total | Accuracy (%) |
|---|---|---|---|---|---|---|---|
| 15 dBm | 3 m | 3.81 | 6.56 | 4.19 | 7.75 | 22.31 | 56 |
| | 5 m | 0.56 | 5.75 | 0.88 | 8.13 | 15.31 | 39 |
| | 7 m | 0.00 | 2.38 | 0.00 | 4.13 | 6.50 | 16 |
| 20 dBm | 3 m | 9.06 | 9.31 | 9.63 | 9.31 | 37.31 | 93 |
| | 5 m | 6.25 | 8.81 | 7.00 | 9.56 | 31.63 | 79 |
| | 7 m | 0.06 | 6.25 | 0.00 | 7.56 | 13.88 | 35 |
| 27 dBm | 3 m | 10.00 | 9.25 | 9.94 | 9.31 | 38.50 | 96 |
| | 5 m | 9.81 | 9.69 | 9.88 | 10.00 | 39.38 | 98 |
| | 7 m | 3.63 | 8.38 | 2.06 | 8.63 | 22.69 | 56 |

**Table 4.** Scanning accuracy averages for plots W7 and W10.

| RFID Pwr | UAV Altitude | L5 | L6 | L8 | L9 | Total | Accuracy (%) |
|---|---|---|---|---|---|---|---|
| 15 dBm | 3 m | 3.03 | 6.07 | 4.01 | 8.42 | 21.53 | 54 |
| | 5 m | 0.36 | 4.96 | 0.77 | 8.35 | 14.45 | 36 |
| | 7 m | 0.00 | 1.81 | 0.00 | 4.85 | 6.67 | 17 |
| 20 dBm | 3 m | 8.32 | 8.91 | 9.40 | 9.57 | 36.20 | 90 |
| | 5 m | 5.42 | 8.28 | 6.79 | 10.11 | 30.60 | 77 |
| | 7 m | 0.41 | 4.83 | 0.04 | 8.03 | 13.31 | 33 |
| 27 dBm | 3 m | 10.00 | 9.17 | 9.97 | 9.66 | 38.79 | 97 |
| | 5 m | 9.78 | 9.51 | 9.85 | 9.96 | 39.10 | 98 |
| | 7 m | 3.56 | 7.90 | 2.95 | 9.10 | 23.51 | 59 |

### 3.6. Statistical Analysis

Linear regression evaluated whether the increasing plant growth affected the total number of tags detected. The parameters for this study's linear regression analysis of interest were the $R^2$ and *standard error*. The $R^2$ determines how close the data points are to the regression line, while standard error measures the average vertical distance between the points and the regression line. In this experiment, the independent variable was the plant growth indices (GIs), while the dependent variables were the total number of tags counted. The plant growth index was linearly associated with the tags counted at RFID power settings of 15 dBm, 20 dBm, and 27 dBm for all UAV flight altitudes from experiments coinciding with plant measurements on September 21, November 1, November 27, and December 14 (excluding plot W7). Table 5 shows the $R^2$ and *p*-values for the different RFID power settings for plots W7 and W10.

**Table 5.** Regression data for plant height and tags counted.

| Plot | RFID Power | Date | Tag Count | GI | $R^2$ | *p*-Value |
|---|---|---|---|---|---|---|
| | | 9/21 | 2550 | 207,001.15 | | |
| | 15 dBm | 11/1 | 2821 | 273,159.75 | 0.31 | 0.62 |
| W7 | | 11/27 | 1900 | 335,375.45 | | |
| | 20 dBm | 9/21 | 6157 | 207,001.15 | 0.81 | 0.28 |
| | | 11/1 | 4144 | 273,159.75 | | |

| | | 11/27 | 2972 | 335,375.45 | | |
|---|---|---|---|---|---|---|
| | 27 dBm | 9/21 | 4673 | 207,001.15 | 0.14 | 0.75 |
| | | 11/1 | 4447 | 273,159.75 | | |
| | | 11/27 | 3076 | 335,375.45 | | |
| | 15 dBm | 9/21 | 2160 | 175,803.92 | 0.37 | 0.39 |
| | | 11/1 | 2102 | 162,659.61 | | |
| | | 11/27 | 2671 | 162,659.61 | | |
| | | 12/14 | 2475 | 166,968.46 | | |
| W10 | 20 dBm | 9/21 | 3499 | 175,803.92 | 0.57 | 0.25 |
| | | 11/1 | 3073 | 162,659.61 | | |
| | | 11/27 | 4729 | 162,659.61 | | |
| | | 12/14 | 4030 | 166,968.46 | | |
| | 27 dBm | 9/21 | 3739 | 175,803.92 | 0.12 | 0.65 |
| | | 11/1 | 4232 | 162,659.61 | | |
| | | 11/27 | 4986 | 162,659.61 | | |
| | | 12/14 | 4497 | 166,968.46 | | |

The $R^2$ values indicated no large negative correlation as hypothesized. The results showed a very small positive linear association and no negative linear association between the variables. This was also supported by the *p*-values, which were greater than 0.05, which shows that the relationship between plant growth and tag count was not statistically significant.

## 4. Discussion

Before the experiment, the hypothesis was that increasing the RFID power and lowering the UAV altitude would provide the highest accuracy in detecting tags and that plant growth would not affect the count accuracy. Tag types L6 and L9 had the highest accuracy, at 100% at 27 dBm and a flight altitude of 3 m. Both tags yielded higher overall accuracy than L5 and L8 in other power settings and altitudes, although L5 and L8 were slightly better in other experimental periods. The L5 and L8 tags were not detected, especially at lower power settings and heights above 3 m, even with two passes. At a higher power setting of 27 dBm and altitudes of 3 m and 5 m, the system had a very high count accuracy, approaching 100% for both plant plots. The negative net difference in the tags detected per pass was lower than 50% at 31% and 22% for plots W7 and W10, respectively, which suggests that a high accuracy level can be sustained by performing only a single pass of the plot.

Plant height and width had no significant increase in the 84-day span (September 21 to December 14). Although the trend in the number of tags detected decreased for the RFID power settings of 20 dBm and 27 dBm, the relationship of the variables was established statistically. The linear regression results suggested that there were no large positive or negative linear associations between the total tags counted and plant growth. What is important to note from the data is that there was no negative value of $R^2$ or large negative linear association for either plot. This finding confirms the hypothesis that plant growth does not affect the scan count of the tags. Furthermore, this also suggests that the plant type/species does not have any significant effect on the RFID signal, as the results of both plots were consistent.

The combination of UAVs and RFID is a novel idea, especially applied to plant inventory, and it is different from image-based systems like cameras, which capture images or reflected light. RFID, on the other hand, receives data by reflecting power on a modulated reflector (tag) [25]. Tags that are within the transmission range transmit binary information to the scanner. The results from the experiments suggest that the use of UAVs and RFID can be a viable solution in plant inventorying, as long the right combination of

factors, such as RFID power and UAV flight level, are met regardless of the type of plant and its growth stage.

## 5. Conclusions

Based on the results from this experiment, the type of tag significantly affects the count accuracy, especially at lower RFID power settings and higher altitudes. The loop-lock tags, L6 and L9, proved to be very efficient at any power setting and altitude, although they had different antenna types. L5 and L8 performed well at the highest RFID power of 27 dBm and lower UAV altitudes of 5 m and below. A second pass was unnecessary since the count accuracy was able to be maintained if the RFID power was set to 27 dBm and the UAV was flown at an altitude of not more than 5 m. On the other hand, plant growth did not affect the tag count. This is important since some nursery plants grow at a higher pace. The more tags read, the better accuracy can be achieved.

Based on the experiments, loop-lock tags for plant inventory in nurseries perform better than stake tags, and this is likely due to signal strength, which is affected by the tag orientation, which is sideways or perpendicular to the plant [23]. For best results with these tags, the RFID power setting should be set to the maximum and the UAV flown below an altitude of 5 m. With only a single pass needed, this can decrease the flight time and reduce battery consumption, therefore covering more production area. As the plants grow, it can be assured that this does not affect the results of the inventory operation.

The next phase of the study will be to integrate the system into the nursery's database and develop an online inventory system. Further studies will be conducted on automating the parts of the procedure to make it robust and practical for operators of this system.

**Author Contributions:** Conceptualization, V.P., J.M.M. and J.R.; methodology, V.P., J.M.M. and J.R.; software, V.P. and J.M.M.; validation, V.P. and J.M.M.; formal analysis, V.P.; investigation, V.P., J.M.M. and J.R.; resources, J.M.M. and J.R.; data curation, V.P.; writing—original draft preparation, V.P, J.M.M. and J.R.; writing—review and editing, V.P., J.M.M. and J.R.; visualization, J.M.M. and V.P.; supervision, J.M.M. and J.R.; project administration, J.M.M.; funding acquisition, J.M.M. All authors have read and agreed to the published version of the manuscript.

**Funding:** This work was partially supported by a grant from Dr. Tanju Karanfil (Vice President of Research) of Clemson University and is based on work supported by NIFA/USDA under project number S1069.

**Data Availability Statement:** The raw data supporting the conclusions of this article will be made available by the authors upon request.

**Acknowledgments:** The authors would like to thank Avery Dennison Corporation and Bennett Dudley of R.A. Dudley Nurseries Inc. for their support and assistance with this project.

**Conflicts of Interest:** The authors declare no conflicts of interest.

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
