# Peer review of "Evaluation of Radio Frequency Identification Power and Unmanned Aerial Vehicle Altitude in Plant Inventory Applications"

_agriengineering, doi:10.3390/agriengineering6020076_

Round 1
Reviewer 1 Report
Comments and Suggestions for Authors
General comment:
The peer-reviewed manuscript is the authors original, innovative research work. The paper has scientific and social justification. I propose that the manuscript be published when the following minor changes and additions are made to it.
Comment 1:
Scientific papers should not be written with subject pronouns (we used; we change, etc.) and possessive adjectives (our hypothesis; and our experiments). The manuscript should be written in the third person and in the past tense.
Comment 2:
The aim of the paper should be stated in the abstract.
Comment 3:
Lines 87-88: The beginning of the sentence should be reworded for better understanding: ”Drones can facilitate the assessment of plant health”.
Comment 4:
Lines 118-119: In the part of the paper that discusses the potential of drones to provide valuable information for human management, the proposal is to note that there is a possibility of communication with a swarm of unmanned aerial vehicles (UAVs) for search purposes during rescue operations. See research: https://doi.org/10.18485/aeletters.2023.8.2.5
Comment 5:
At the beginning of the chapter - 2. Materials and method, in one sentence it should be pointed out which methods, techniques, analyzes and software were used in the research.
Comment 6:
Lines 247-249: Need to separate units from numbers.
Comment 7:
Lines 279-280: This text should be given in Chapter 2. Materials and Method.
Comment 8:
What could be the continuation of the research should be pointed out in the conclusion.
Comment 9:
The Discussion and Conclusion are very well written.
Comment 10:
The choice of choruses is good. It is recommended to include several recent researches in order to enrich the manuscript.
Author Response
Comments and Suggestions:
- Comment 1: Scientific papers should not be written with subject pronouns (we used; we change, etc.) and possessive adjectives (our hypothesis; and our experiments). The manuscript should be written in the third person and in the past tense.
This has been addressed by revising the text and removing the subject pronouns and possessive adjectives. Thank you for the suggestion.
- Comment 2: The aim of the paper should be stated in the abstract.
We appreciate the reviewer pointing this out. The aim of the study has been reflected in the abstract. (lines 31-34).
- Comment 3: The beginning of the sentence should be reworded for better understanding: ”Drones can facilitate the assessment of plant health”.
The recommended rewording has been carried out. Thank you for the comment. (Lines 89-90)
- Comment 4: Lines 118-119: In the part of the paper that discusses the potential of drones to provide valuable information for human management, the proposal is to note that there is a possibility of communication with a swarm of unmanned aerial vehicles (UAVs) for search purposes during rescue operations. See research: https://doi.org/10.18485/aeletters.2023.8.2.5
We appreciate your suggestion to add that paper. Drone swarm could increase the efficiency of plant inventory by not only relying on a single drone. (lines 129-131)
- Comment 5: At the beginning of the chapter - 2. Materials and method, in one sentence it should be pointed out which methods, techniques, analyzes and software were used in the research.
Thank you very much for the comment. A sentence was added in the beginning of Materials and Methods describing the methods and techniques used in the study. (Lines 139-140)
- Comment 6: Lines 247-249: Need to separate units from numbers.
We updated the manuscript. (Lines 269-279)
- Comment 7: Lines 279-280: This text should be given in Chapter 2. Materials and Method.
We appreciate your valuable comment and updated the materials and methods. (Lines 231-234)
- Comment 8: What could be the continuation of the research should be pointed out in the conclusion.
We appreciate very much your comment on this, and we gladly carried this out by adding the future work in the conclusion. (lines 453-456)
- Comment 9: The Discussion and Conclusion are very well written.
Thank you very and we appreciate sharing your expertise in reviewing the paper.

Reviewer 2 Report
Comments and Suggestions for Authors
- In abstract, the authors used “Field data were collected on September 9, 21-22, October 19, November 1-2 and 27, and December 14, 2023 (total of 8 dates)”. What is this representation? More over exact date are not required in the abstract.
- At the end of introduction section, it is required to discuss the limitations and the research gap present in the current situation.
- The “The UAV can handle a maximum payload of 6 kg, including a single Turnigy 2200 mAh LiPo battery” is not correct. There are other UAVs there with higher pay load. Please check where this pay load has been used.
- The Table 1 deals with summary of RFID tag types. What type of tag it is? Is it active or passive? How much distance it could cover? What is the frequency range?
- It is required to mention the latitude and longitude related to Figure 6. W7 study area (a) and RFID tag assignment including the marker tags (b).What is the maximum distance between UAV and plant?
- Why the UAV has to be operated 3 m, 5 m and 7 m power rating?
- Rearrangement / additional detailed information are needed in Figure 8. Data processing flow.
- Statistical Analysis is not enough. More detailed information are needed.
- In Figure 10. Average of unique tags detected per tag type for plot W7 and Figure 11. Average of unique tags detected per tag type for plot W10 are strted from L5?
- Why negative tag count is present in Figure 13. Net difference of the tags scanned after the second pass for plot W10 and in fig 12 as well. Projet this figure in alternative way if there is no negative tag count.
- In results and discussion, the statics are clearly given by the author.
112. There is no much information related to future in the conclusion.
- Additional recent references are required to support the present research. The present numbers are not sufficient.
Moderate editing of English language required
Author Response
Comments and Suggestions:
- Comment 1: In abstract, the authors used “Field data were collected on September 9, 21-22, October 19, November 1-2 and 27, and December 14, 2023 (total of 8 dates)”. What is this representation? More over exact date are not required in the abstract.
We appreciate your comment on this. The date represents that our experiments were done over a span of 4 months, and that is, there were no uniform intervals. We gladly carried out your recommendation by narrowing down the text to state only the start month and end month. (line 24)
- Comment 2: At the end of introduction section, it is required to discuss the limitations and the research gap present in the current situation.
Thank you very much for pointing this out. We gladly carried out this recommendation by adding texts discussing the current situation's limitations and research gaps. (lines 132-137).
- Comment 3: The “The UAV can handle a maximum payload of 6 kg, including a single Turnigy 2200 mAh LiPo battery” is not correct. There are other UAVs there with higher pay load. Please check where this pay load has been used.
We appreciate very much your comment on the maximum payload. However, according to the manufacturer’s website, it can handle a maximum of 6kg when using 6 TB47S batteries (click here for the specification; see page 7). During the experiment, we were using TB47S batteries. We just stated the maximum payload of the UAV, but our payload is much less than that. We hope that this address your concern on maximum payload. Also the Turnigy battery do not have substantial weight as it only has 2,200 mAh.
- Comment 4: The Table 1 deals with summary of RFID tag types. What type of tag it is? Is it active or passive? How much distance it could cover? What is the frequency range?
Thank you for the comment. We acknowledge that we failed to include these details. The tag types used were passive, operating at the frequency of 900 MHz. The manuscript was revised to reflect your recommendations. (lines 150-151).
- Comment 5: It is required to mention the latitude and longitude related to Figure 6. W7 study area (a) and RFID tag assignment including the marker tags (b).What is the maximum distance between UAV and plant?
We appreciate your recommendation very much. We added the latitude and longitude of plots W7 and W10 (lines 203-204). The RFID tag assignment and marker tags are shown in Figure 6. We did not get the exact location for each tag in GPS format as we used the nursery placement on the plants and feel that the location for each tag may be less important as they are too near from each other.
- Comment 6: Why the UAV has to be operated 3 m, 5 m and 7 m power rating?
One objective was to investigate the effect of drone altitude on tag counts. The altitudes selected for this experiment were 3m, 5m, and 7m. Our prior work related to this experiment and usage has indicated that these altitudes will give us data that would be more applicable to RFID reader. Also, flying less than 3m would not be conducive for a nursery as there were plants that could reach that height.
- Comment 7: Rearrangement / additional detailed information are needed in Figure 8. Data processing flow.
Thank you for pointing this out to provide clarity to Figure 8. We have added additional texts to explain the data processing flow further. (line 250-255)
- Comment 8: Statistical Analysis is not enough. More detailed information are needed.
Thank you very much for this important comment. We have added more to the statistical analyses used. (line 293-301)
- Comment 9: In Figure 10. Average of unique tags detected per tag type for plot W7 and Figure 11. Average of unique tags detected per tag type for plot W10 are strted from L5?
We appreciate very much this question for clarification. Figure 10 represents the unique tag counts, which are the average of all the experiment data we gathered. We classify the tag counts into their tag counts. For example, in tag type L5 in plot W10, the count is about 4 (out of 10) at 15 dBm at 3 m flight level. When the flight level was increased, the figure shows the average is less than one. This can be explained since, in some of our experiments, at the same power setting and flight level, tag type L5 was not detected at all.
- Comment 10: Why negative tag count is present in Figure 13. Net difference of the tags scanned after the second pass for plot W10 and in fig 12 as well. Project this figure in alternative way if there is no negative tag count.
We appreciate your comment on this part of the paper. We clarified that the net difference represents the “count loss” on the second pass. We want to show how many unique tag counts where not counted in the second pass compared to the first pass. We have updated the discussion on Figure 13 to explain it further. (line 339-344)
- Comment 11: In results and discussion, the statics are clearly given by the author.
We apologize but do not know what Reviewer #2 refers to by this statement.
- Comment 12: There is no much information related to future in the conclusion.
We appreciate very much your comment on this, and we gladly carried this out by adding the future work in the conclusion. (lines 463-464)

Reviewer 3 Report
Comments and Suggestions for Authors
The study performed on the application of Radio-Frequency Identification (RFID) and Unmanned Aerial Vehicles (UAVs) technologies to plant inventory is interesting and properly done. The manuscript is properly organized and the presentation of the research activity is adequately. However, the presence of some minor aspects to be solved makes the manuscript not yet ready for publication. More in details:
1 - The sentence at page 5 rows 163-164 is a repetition of that at the same page and rows 171-172.
2 - To increase clarity in reading Figure 3, it would be preferable to add some explanatory text of the various components that make up the UAV-RFID detection system.
3 - Figure 4 without any comment in the text on the dashboard functionality is unuseful.
4 - It is required to add details in the figure caption and/or comment in the text on figure 6.b for clarity.
5 - It is required comment on the data processing performed and shown as block diagram in figure 8.
6 - Comment about results shown in figure 10 and 11 is required. How does the position of the RFID tag on the plant, in the case of the loop-look attachment (L6 and L9), influence the ability of the UAV-RFID system to detect the tag itself? However, in the case of RIFD tags inserted into the plant pot, how can the conformation of the plant influence the ability to detect the tags themselves (stake attachment, L5 and L8)?
7 - It is necessary to comment on the results shown in Figures 112 and 13. Are the negative differences obtained a consequence of some communication error?
8 - Comment is needed on the results shown in Figure 14. It is unclear the decreasing trend seen over time on the tag count, and the lower tag count at 27 dBm power compared to 20 dBm recorded in different situations.
9 - Sentences at page 11 from row 315 to row 322 are not clear and not supported by data.
10 - Sentences at page 11 from row 336 to row 344: check the days on which the tests were carried out, there appear to be some inconsistencies with what is indicated in other sections of the manuscript.
11 - Discussion at page 13 need to be improved with comment on the experimental results to better highlight the work performed and its level of novelty.
Author Response
Comments and Suggestions:
- Comment 1: The sentence at page 5 rows 163-164 is a repetition of that at the same page and rows 171-172.
The sentence in line 171-172 was removed. We appreciate the suggestion.
- Comment 2: To increase clarity in reading Figure 3, it would be preferable to add some explanatory text of the various components that make up the UAV-RFID detection system.
We appreciate the suggestion. The components of the UAV have been explained, and some texts have been added to provide more details. (lines 183-185).
- Comment 3: Figure 4 without any comment in the text on the dashboard functionality is unuseful.
The suggestion was carried out with an additional explanation of the functions of both applications. Thank you very much for pointing it out. (lines 189-196)
- Comment 4: It is required to add details in the figure caption and/or comment in the text on figure 6.b for clarity.
The suggestion was carried out by additional text describing Figure 6b in lines 210-214. Thank you for the suggestion.
- Comment 5: It is required comment on the data processing performed and shown as block diagram in figure 8.
The suggestion was carried out by an additional paragraph commenting on the process shown in Figure 8 (lines 250-255).
- Comment 6: Comment about results shown in figure 10 and 11 is required. How does the position of the RFID tag on the plant, in the case of the loop-look attachment (L6 and L9), influence the ability of the UAV-RFID system to detect the tag itself? However, in the case of RIFD tags inserted into the plant pot, how can the conformation of the plant influence the ability to detect the tags themselves (stake attachment, L5 and L8)?
We appreciate this comment, we included an additional paragraph explaining the performance of the different tag types by citing the previous study on the same tags used (lines 326-328).
- Comment 7: It is necessary to comment on the results shown in Figures 112 and 13. Are the negative differences obtained a consequence of some communication error?
In the text, we clarified that the net difference represents the “count loss” on the second pass (lines 339-344). We want to show how many unique tags were not counted in the second pass compared to the first pass. The negative difference is not a consequence of communication error.
- Comment 8: Comment is needed on the results shown in Figure 14. It is unclear the decreasing trend seen over time on the tag count, and the lower tag count at 27 dBm power compared to 20 dBm recorded in different situations.
We addressed this comment by including an additional text explaining the decline of the tag counts at power levels 20 dBm and 27 dBm (lines 358-362).
- Comment 9: Sentences at page 11 from row 315 to row 322 are not clear and not supported by data.
We appreciate your comment for increased clarity, so we have re-written those specific sentences. At the highest power level for Plot W7 the trend shows that the number of tags read decreased until the final day. This unexpected increase in counts for plot W7 on the final day cannot be explained. For both plots the trends suggest that there was no increase in the number of tags counted over the entire experimental period. (line 358-362)
- Comment 10: Sentences at page 11 from row 336 to row 344: check the days on which the tests were carried out, there appear to be some inconsistencies with what is indicated in other sections of the manuscript.
We corrected the manuscript to remove September 9 as we did not measure the plant at that date. (line 378-379)
- Comment 11: Discussion at page 13 need to be improved with comment on the experimental results to better highlight the work performed and its level of novelty.
We appreciate very much your comment on the need to improve the discussion to better highlight the work performed and its level of novelty. We gladly revised the manuscript to make the improvements needed. (line 420-426)

Round 2
Reviewer 2 Report
Comments and Suggestions for Authors
Authors have addressed all the points represented by the reviewer